# Newcastle Disease Genotype VII Prevalence in Poultry and Wild Birds in Egypt

**DOI:** 10.3390/v14102244

**Published:** 2022-10-13

**Authors:** Amal A. M. Eid, Ashraf Hussein, Ola Hassanin, Reham M. Elbakrey, Rebecca Daines, Jean-Remy Sadeyen, Hanan M. F. Abdien, Klaudia Chrzastek, Munir Iqbal

**Affiliations:** 1Department of Avian and Rabbit Medicine, Faculty of Veterinary Medicine, Zagazig University, Zagazig 44511, Egypt; 2The Pirbright Institute, Ash Road, Pirbright, Woking GU24 0NF, UK; 3Pathobiology and Population Sciences, Royal Veterinary College, Hatfield AL9 7TA, UK; 4Department of Avian and Rabbit Medicine, Faculty of Veterinary Medicine, Suez Canal University, Ismailia 41622, Egypt

**Keywords:** migratory birds, Newcastle disease virus-GVII, poultry, phylogenetics, sequence-independent, single-primer amplification (SISPA), velogenic, whole genome sequencing (WGS)

## Abstract

Newcastle Disease Virus (NDV) genotype VII is a highly pathogenic *Orthoavulavirus* that has caused multiple outbreaks among poultry in Egypt since 2011. This study aimed to observe the prevalence and genetic diversity of NDV prevailing in domestic and wild birds in Egyptian governorates. A total of 37 oropharyngeal swabs from wild birds and 101 swabs from domestic bird flocks including chickens, ducks, turkeys, and pelicans, were collected from different geographic regions within 13 governorates during 2019–2020. Virus isolation and propagation via embryonated eggs revealed 91 swab samples produced allantoic fluid containing haemagglutination activity, suggestive of virus presence. The use of RT-PCR targeted to the F gene successfully detected NDV in 85 samples. The geographical prevalence of NDV was isolated in 12 governorates in domestic birds, migratory, and non-migratory wild birds. Following whole genome sequencing, we assembled six NDV genome sequences (70–99% of genome coverage), including five full F gene sequences. All NDV strains carried high virulence, with phylogenetic analysis revealing that the strains belonged to class II within genotype VII.1.1. The genetically similar yet geographically distinct virulent NDV isolates in poultry and a wild bird may allude to an external role contributing to the dissemination of NDV in poultry populations across Egypt. One such contribution may be the migratory behaviour of wild birds; however further investigation must be implemented to support the findings of this study. Additionally, continued genomic surveillance in both wild birds and poultry would be necessary for monitoring NDV dissemination and genetic diversification across Egypt, with the aim of controlling the disease and protecting poultry production.

## 1. Introduction

Poultry in Egypt has become the substantial source of animal protein. Such a huge industry is challenged by several devastating pathogens. Newcastle Disease (ND) is one of the endemic diseases that still subtract from the outcome of poultry through continuous spread among poultry flocks, even within vaccinated populations. Avian Orthoavulavirus 1 (AOaV-1), also known as Newcastle Disease Virus (NDV), is an enveloped negative-sense, single-stranded RNA virus belonging to the family *Paramyxoviridae*, order *Mononegavirales* [1]. The viral genome is around 15,200 base pairs (bp) in length and encodes six different proteins: nucleocapsid protein (NP); phosphoprotein (P); matrix protein (M); large RNA polymerase (L); fusion protein (F); and haemagglutinin–neuraminidase (HN). Two other proteins (V and W) could also be coded through P protein mRNA editing [2,3]. The phylogenetic analysis of F gene sequences of NDVs divided them into two classes (I and II): class I includes avirulent viruses, with a natural reservoir of aquatic wild birds, but one virulent isolate has been included [4]; whereas class II contains viruses that have higher genetic and virulence variability with at least 20 genotypes (I–XXI, except the recombinant sequence genotype XV) and are known to infect a wide range of domestic and wild birds [2,3,5].

ND was first identified in Egypt in 1948 [6], with genotype II (NDV-II) being most prevalent. However, since 2011, reported outbreaks revealed a new commonly occurring genotype, NDV-GVII, with a velogenic pathotype and varying incidence among vaccinated flocks of commercial and backyard chickens throughout the Egyptian governorates [7,8,9,10,11,12].

More than 250 domestic and wild bird species have been shown to be susceptible to NDV infection with varying clinical forms [13,14,15,16]. However, virus transmission among wild birds possesses serious risk for both poultry and other wild bird populations [17,18]; virus adaption can encourage new susceptible host species, posing a greater threat to local as well as wider geographical regions [17,18,19]. Mansour et al. [20] reviewed the clinical disease, epidemiology, and evolutionary perspective, of NDV among domesticated and wild birds in the view of available research work publications. The virulence of NDV strains varies significantly between hosts and to date, ducks are less susceptible to NDV infection [21]; NDV strains of varying virulence and pathogenicity have been isolated from either diseased or apparently healthy ducks, which questions whether ducks are natural reservoirs or less susceptible hosts to NDV [22,23].

The distinctive intermixing of migratory, free-living, and domesticated birds in Egypt might facilitate the direct transmission to commercial flocks and backyard populations, besides the evolution power of APMV-1 mutation [20]. Egypt boasts a crucial geographical location for migratory birds, bridging three continents and hosting an essential resting place for those migrating from Europe in the spring and autumn months. It is also considered the second most important migratory pathway for birds in the world, with more than two million birds passing through annually [24]. Despite extensive vaccination regimens across commercial and backyard poultry flocks, Egypt has and remains to witness devastating outbreaks, resulting in huge socio-economic losses [20].

The frequency of virulent strains reported in wild birds on other continents is also increasing [25,26,27], which questions the possibility that wild birds play a potential role in the spread of virulent NDV in Africa, especially in Egypt. Additionally, low biosecurity measures could endanger the poultry industry, increasing the possibility of disease transmission through wildlife-poultry contact or contaminated fomites [28]. However, there are few reports of ND virus isolation in epizootiological surveys of wild birds, with limited sequences and availability of data [29,30]. Therefore, the role of these birds in the maintenance of the disease is still obscure.

The available research investigating the dissemination and diversity of NDV across Egypt is limited, yet necessary for strategic planning for ND control. In this present study, we investigated the prevalence of NDV in domestic, migratory, and non-migratory birds, evaluating the genetic diversity and present genotypes within geographically distant governates in Egypt.

## 2. Materials and Methods

### 2.1. Study Area and Sample Collection

The primary aim of this study was to monitor the genotypic diversity of NDV prevailing in domesticated and wild birds (migratory and non-migratory) in Egypt. The active surveillance of wild birds is difficult due to practical, logistical, and financial constraints; thus, we defined clear strategies to obtain samples that would provide us with robust sources of targeted NDV prevalence. Therefore, samples were mainly collected from domestic birds (semi-intensive housing system) that could potentially serve as “sentinels”, and from wild birds that could serve as a reservoir for virus persistence and transmission to poultry.

The study was performed from September 2019 to December 2020 across different governorates (n = 13) within Egypt, including Dakahlia, Sharkia, Cairo, Suez, Sohag, Asyut, Port-Saied, North Sinai, South Sinai, Alexandria, Ismailia, Damietta, and Gharbia. Oropharyngeal swabs were collected from 101 flocks of domestic birds and 37 from migratory and non-migratory wild birds (total n = 138).

#### 2.1.1. Domestic Birds

Oropharyngeal swabs collected from the 101 flocks of domestic birds (pool of 3 swabs/flock) included chickens (n = 60), ducks (n = 38), pelicans (n = 2), and turkey (n = 1), of different ages and breeds (Table 1). At the time of the farm visit, samples were collected from both apparently healthy and diseased birds. The sick birds that were either dead or diseased were clinically examined and post-mortem observations were recorded in addition to the mortality rates. Only chicken and turkey flocks received vaccination with live and inactivated NDV vaccines, however, no vaccination regimens for NDV were administered to the wild birds, ducks, and pelicans.

#### 2.1.2. Migratory and Non-Migratory Wild Birds

A total of 37 apparently healthy wild birds, including 14 different species, were captured randomly via traps, or collected from live-bird-markets (LBMs) in different governorates including North Sinai, Dakahlia, Port-Said, Ismailia, and Sharkia, Egypt (Table 2). The traps used to capture were baited nets that were erected at the entrances to the lakes at Al Manzala Lake in Port-Said, Bardawil Lake in North Sinai, and Bitter Lakes near to Ismailia. The birds caught were classified and scientifically named according to species, order, and breeds, by the wildlife scientist at the Faculty of Veterinary Medicine-Suez Canal University. Oropharyngeal swabs were collected largely during the autumn and spring seasons. Sterile cotton swabs (propylene stick) were used to collect samples from the oral cavity of birds in cryovials with virus transport media (5% glycerol-MEM media pH 7.2 supplemented with 1000 IU of penicillin and 1000 µg streptomycin per mL) [31]. The samples were kept refrigerated (at 4 °C) and transported to the laboratory of the Department of Avian and Rabbit Medicine, Faculty of Veterinary Medicine, Zagazig University, Egypt, within 24 h and stored at −80 °C until further processing.

### 2.2. Sample Processing and Virus Isolation

The oropharyngeal swab samples were centrifuged at 3000 rpm for 10 min, and the supernatants were filtered using 0.22 µm sterile filter. After aseptic examination, 200 µL of filtrate was inoculated to 10 day-old specific pathogen-free (SPF) embryonated chicken eggs (ECEs) via the allantoic cavity using a standard egg inoculation procedure [31]. Three fertile eggs per test or negative control sample were used. The inoculated eggs were incubated at 37 °C for a maximum of 5 days with daily monitoring. Embryos that were found dead within the first 24 h post-inoculation were regarded as non-related unspecific death and were discarded. After 24 h, any embryo that showed typical signs of virus infection such as haemorrhage, blood vessels coming away from the shell, and/or no movement of the embryo during the candling process, were immediately chilled at 4 °C. The embryos that remained live until 5 days post-inoculation (pi), where they were chilled at 4 °C for a minimum of 4 h. Allantoic fluids (AFs) were harvested and subjected to rapid haemagglutination assay (HA) by using 1% (*v*/*v*) washed chicken red blood cells (RBCs). Allantoic fluids with positive HA activity were subjected to RT-PCR for NDV detection and sequencing.

### 2.3. RNA Extraction and Reverse Transcriptase-Polymerase Chain Reaction (RT-PCR)

Viral RNA was extracted either directly from oropharyngeal swabs or HA positive allantoic fluid using the QIAamp MinElute Virus Spin kit (Qiagen GmbH, Hilden, Germany, catalogue no. 57704) following the manufacturer’s guidelines. As positive and negative control samples, the NDV vaccine strain (La Sota) and allantoic fluid from non-infected embryos were used, respectively. The extracted RNAs were reverse transcribed to cDNA using random primers and the assay was processed with the High-Capacity cDNA Reverse Transcription Kit (Applied Biosystems^TM^, Loughborough, UK, catalogue no. 4268814) following the manufacturer’s instructions.

The PCR amplification was performed using primers targeting the fusion (F) gene of NDV (forward: 5′-CACAGCAGGTCGGTGTAGAA-3′ and reverse: 5′-TCTCCAAATAGGTGGCACGC-3′) and conducted using 2X Dream^Taq^ Green PCR Master Mix (ThermoFisher Scientific, Loughborough, UK, catalogue no. K1081). Cycling conditions were performed as a single cycle of initial denaturation at 95 °C for 3 min, followed by 30 cycles of 94 °C for 30 s (denaturation), 60 °C for 30 s (annealing), and 72 °C for 45 s (extension), then 72 °C for 7 min of final extension step. After amplification, the PCR products were resolved on a 1% agarose gel and stained with ethidium bromide; positive amplification products yielded a band size of approximately 306 bp. Additionally, the positive HA allantoic fluids were screened for the presence of influenza A virus using the primers targeting the conserved region of influenza’s M gene [32].

### 2.4. Sequence-Independent Single Primer Amplification (SISPA)

Extracted RNA was used in SISPA reaction as described previously [33]. Briefly, for cDNA: 1 μL of 100 μM primer K-8N [33] and dNTPs (10 μM each) were used in 20 μL reaction mixture. Reverse transcription (RT) was performed with SuperScript IV Reverse Transcriptase (ThermoFisher Scientific) at 55 °C for 10 min, followed by 80 °C for 10 min and then placed on ice. After RT-PCR, 20 μL of first-stranded cDNA was heated at 94 °C for 3 min, then chilled on ice for 3 min with 10 μM of primer K-8N (0.5 μL per reaction), and 10 μM dNTPs (0.5 μL per reaction) in 1× Klenow reaction buffer ((New England Biolabs, Hitchin, UK (NEB)) [33]. 1 μL of Klenow fragment (NEB) was added and the reaction was incubated at 37 °C for 60 min. the resulting dsDNA was cleaned using Agencourt AMPure XP beads (Beckman Coulter, CA, USA); purified dsDNA was subsequently used as a template for PCR amplification. 5 μL of purified dsDNA was taken forward for sequence independent PCR amplification, containing 1x Q5 High-Fidelity Master Mix (NEB), 2.5 μL of 10 μM primer K (5′-GACCATCTAGCGACCTCCAC-3`) and nuclease-free water with a final reaction volume of 50 μL. The PCR cycling conditions were as follows: 98 °C for 30 s, followed by 35 cycles of 98 °C for 10 s, 55 °C for 30 s, and 72 °C for 1 min, with a final extension at 72 °C for 10 min; PCR products were purified using Agencourt AMPure XP beads (Beckman Coulter). Obtained dsDNA was quantified by Quibt dsDNA HS assay (Invitrogen, Loughborough, UK) following manufacturer’s instruction and purified dsDNA was subsequently used for genome sequencing.

### 2.5. Genome Sequencing

A Nextera XT DNA kit (Illumina, San Diego, CA, USA) was used following manufacturer’s instructions to generate multiplexed paired-end sequencing libraries from 1 ng of dsDNA, using methods as previously described [33]. Libraries were analysed on a High Sensitivity DNA Chip on the Bioanalyzer (Agilent Technologies), pooled, and sequenced on a 2 × 300 cycle using the MiSeq Reagent Kit v2 (Illumina, San Diego, CA, USA).

### 2.6. Whole Genome Sequencing Data Analysis

The raw sequencing reads were analysed as described previously [33,34]. Briefly, the quality of reads was assessed using the FastQC ((version 0.11.5) Andrews. S, Cambridge, UK) and were quality trimmed (employing a quality score of ≥30) with adaptor removal using the Trim Galore ((version 0.5.0) Krueger. F, Cambridge, UK) (https://github.com/FelixKrueger/TrimGalore, accessed on 5 February 2019) [35]. De novo assembly was performed using the SPAdes de novo assembler ((version 3.10.1) University of California, San Diego, CA, USA) (k-mer 33, 55, and 77) with resulting contigs quality assessed using the QUAST ((version 5.0.2) Gurevich et al., St Petersburg Academic University, Saint Petersburg, Russia) [36,37]. Reference-based orientation and scaffolding were performed using the Scaffold_builder ((version 2.2) Silvia. G. G, San Diego, CA, USA) [38]. Consensus sequences were assigned based upon BWA-MEM mapping of trimmed (but un-normalized) read data to the genome scaffold and parsing of the mpileup alignment, the BWA-MEM ((version 0.7.17) Li. H, Durbin. R, Cambridge, UK) and the Geneious ((version 9.1.2) Dotmatics, MA, USA) (https://www.geneious.com, accessed on 5 February 2019) were used to reference genomes [39].

### 2.7. Phylogenetic Analysis

Complete coding sequences of F gene (n = 5) were used for comparative genetic analyses. To determine the genotypes of NDV viruses, we used previously published NDV reference strains [3] and the maximum likelihood (ML) phylogenetic tree of the F segment was constructed using nucleotide sequences. These sequences were aligned using MUSCLE [40] and ML phylogeny was generated using GTR nucleotide substitution model, with among-site rate variation modelled using a discrete gamma distribution (GTR + G) using the MEGA 11 ((version 11.0.8) Tamura et al., Tokyo, Japan); ML model selection prior to tree construction recommended such model parameters incurring the lowest Bayesian Information Criterion (BIC) [41]. Bootstrap support values were generated using 500 rapid bootstrap replicates.

### 2.8. Pathogenicity Indices

The pathogenicity of the six sequenced NDV strains were assessed using mean death time (MDT) and intra cerebral pathogenicity index (ICPI). For the calculation of MDT, 10-fold serial dilutions (10^−1^ to 10^−9^) were prepared from fresh infective allantoic fluid using a sterile phosphate-buffered saline. 0.1 mL from each dilution was inoculated through the allantoic sac route of five SPF ECEs at 9–10 days old. The inoculated eggs were incubated at 37 °C and examined daily for 5–7 days, recording rate of mortality. The embryonic MDT was determined as described by Alexander and Senne [15].

For the calculation of ICPI, 60 one-day old chicks (10 chicks/sequenced NDV strain) were intracerebrally injected with a volume of 50 µL of a 10-fold dilution, prepared from freshly allantoic fluid of each NDV strain. The inoculated chicks were monitored daily for 8 days post-inoculation. The observation welfare scoring the chicks’ observation and the value of ICPI was determined according to OIE [31].

## 3. Results

### 3.1. Clinical Signs and Post-Mortem Lesions

At the time of sampling, both migratory and non-migratory wild birds were apparently healthy with no clinical disease signs. However, the clinical disease signs were observed in the domestic birds. In chickens, ruffled feathers, greenish diarrhoea, and respiratory distress (sneezing, swollen eyes, and gasping) were the predominant signs. Cyanosis of the head, comb, and wattles, as well as nervous distress, was also observed in some chickens; while in ducks, the predominant clinical signs were nervous distress and greenish diarrhoea. Furthermore, respiratory distress and diarrhoea were apparent in the turkey flock. The post-mortem examination of euthanized birds showed septicaemia and congestion in the internal organs and enteritis, with greenish intestinal contents in all examined birds. Haemorrhages were also observed in the caecal tonsils, the mucosa, and the glands, of the proventriculus of some chickens.

### 3.2. Virus Identification

The inoculation of SPF-ECEs with swab samples showed that some embryos died within 96 h. Among the infected embryos, 80.3% showed diffuse haemorrhages and congestion suggesting possible virus presence, either NDV and/or AIV. The tested harvested allantoic fluids (n = 138) revealed positive HA activity in 65.9% of samples (91/138). The positive HA samples were further confirmed by RT-PCR; NDV was detected in 93.4% (85/91) of the positive HA samples, which corresponded to 61.6% (85/138) of the total collected samples; 42% (58/138) were from domestic poultry flocks and 19.6% (27/138) were from wild birds. Finally, the total positive NDV samples of each population were 57.4% (58/101) and 72.9% (27/37) for domestic poultry and wild birds, respectively. From the 85 samples positive for NDV, 83.5% (71/85) were associated with a single infection with NDV, of which 71.8% (51/71) of these samples were from domestic poultry and 28.2% (20/71) wild birds, respectively. These corresponded to 51.4% (71/138) of total combined samples, 50.5% (51/101) of domestic poultry samples, and 54.1% (20/37) of wild bird samples, respectively. Further analysis showed the presence of AIV in 14.5% (20/138) of samples, either as a combined infection of both AIV and NDV 70% (14/20) from both domestic and wild bird samples (7 positive per each) or AIV alone in 30% (6/20) samples collected from domestic birds (Figure 1 and Figure 2).

### 3.3. Incidence of NDV Based on F Gene

The isolation of NDV in the different governorates (n = 13) of Egypt and the case number analysed (n/n) are shown in Figure 3. Among the varied species of examined domestic birds, 65% (39/60) of chicken flocks (95% CI 51.60–76.87%) and 47.4% (18/38) of the duck flocks were positive for NDV infection (95% CI 30.97–64.19%). The single turkey flock assessed was also positive for NDV, but pelican flocks revealed no virus detection (Table 3). Whereas in the wild birds, the NDV infection rate of the species *Larus canus*, *Numenius minutus*, *Streptopelia turtur, Gallinula chloropus*, *Ardenna pacifica*, *Porphyrio madagascariensis*, and *Anthus rubescens* reached 100%, followed by *Corvus cornix* with an incidence of 75% (6/8) (95% CI 34.91–96.61%) (Table 4). Six isolates were selected at random as representative strains for sequence analysis by whole genome sequencing (WGS); detailed information shown in Table 5.

### 3.4. Phylogenetics

Only five full F gene sequences were included in phylogenetic analysis (Table 5). Among the five NDV isolates analysed, three isolates were derived from chickens, one domestic duck, and a black-crowned night heron, in Egypt in 2019. The homology analysis of the five viral genome sequences demonstrated that the F genes shared very high nucleotide sequence identity (98.014–100%) (Appendix A). Two chicken isolates NDV/Chicken/Egypt/NOR/ZU-NM76/2019 and NDV/Chicken/Egypt/ALEX/ZU-NM99/2019 clustered together with NDV/Black-crowned_night_heron/Egypt/POR/ZU-NM85/2019, a wild bird isolate, whereas the remaining chicken isolate, NDV/Chicken/Egypt/ALEX/ZU-NM97/2019, clustered together with mallard duck isolate NDV/Duck/Egypt/DAK/ZU-NM09/2019 (Appendix A). In addition, homology analysis was also performed using the whole NDV genome sequences obtained in this study. Two chicken and one duck isolates (99% genome coverage), along with black-crowned night heron isolate (90%), were included in this analysis (Table 5). High identity (98.63%) was found between NDV/Chicken/Egypt/NOR/ZU-NM76/2019 and NDV/Chicken/Egypt/ALEX/ZU-NM99/2019, followed by 96.87–97.18% of nucleotide sequence identity between the chicken isolates and duck isolate, NDV/Duck/Egypt/DAK/ZU-NM09/2019 (Appendix A). Although we did not assemble a complete genome sequence of wild bird isolate (NDV/Black-crowned_night heron/Egypt/POR/ZU-NM85/2019), the homology analysis showed 95.63–96.19% identity with two chicken isolates and 94.48% nucleotide identity with the duck isolate (Appendix A).

All isolates belonged to class II and clustered with NDV isolates within genotype VII.1.1 (Appendix A). The phylogenetic analysis of other representative isolates of genotype VII.1.1 strains shared a high F-gene nucleotide similarity (97.83–99.49%) with those isolated in Egypt between 2012–2016 (Appendix A and Appendix A). The isolates also share high similarity with 2017 genotype VII Israeli strains and 2011 Chinese isolates (97.95–98.01% similarity). The F protein cleavage site of the sequence isolates contained polybasic residues (RRQXRF), where X is R or K pathotype. High virulence was confirmed with MDT and ICPI values of 35.8–64 h and 0.98–1.670, respectively. ICPI values ≥ 0.7 are deemed virulent and coupled with the presence of a multi-basic cleavage site, confirms pathogenicity.

The comparative alignment of the complete F protein amino acid sequence of the isolated strains against each other revealed four frequent residue substitutions (N/S, T/A, N/S, and R/K) at positions, 30, 90, 258, and 480, respectively, resulting in independent sequences (Appendix A).

## 4. Discussion

The global distribution of NDV and epidemiological analysis portrays genotype VII as the predominant genotype and is responsible for the fourth major NDV panzootic worldwide, with often fatal consequences following the infection of susceptible birds [42,43]. The poultry industry in Egypt is equally suffering massive economic losses due to NDV-GVII infections, despite delivering mass vaccination schedules to all commercial flocks. As such, this study sought to develop the epidemiological picture and dissemination of NDV in domestic and wild bird populations throughout 13 governorates of Egypt.

The incidences of NDV among domestic birds were 65% and 47.4% in chicken and duck flocks, respectively; similar levels of NDV prevalence in domestic poultry (52.4%) were found in a cohort epidemiological study by Abozaid et al. [44]. The observed higher prevalence of NDV in wild birds (72.9%) compared to domestic birds (57.4%) was surprising. A similarly high prevalence of NDV infection in wild birds (60%) was recorded by Ameji et al. [45]. Conversely, a low detection rate of NDV was recorded in wild birds with a percentage of 5.18% [46]; 3.06% [47], and 3.77% [29]. Among the 85 samples positive for NDV, the majority 71 (61.6%) contained NDV alone, but 14 samples (38.4%) contained both NDV and AIV. Such incidences of NDV and AIV co-infections are not rare and have been widely reported in many regions of Egypt, with higher rates of occurrence where both pathogens are enzootic poultry and wild birds [8].

Previous studies suggest that wild birds may act not just as a reservoir of low virulence strains but may also play a critical role in the epidemiology of different variants NDV strains persisting in Africa, including virulent Newcastle Disease Virus (VNDV) strains responsible for poultry outbreaks [17,47,48,49,50]. Therefore, we anticipated that samples from wild birds and domestic birds may mostly contain low virulence strains. Contradictory to this hypothesis, all NDV positive wild bird and domestic poultry samples were VNDV-GVII, confirmed by the presence of polybasic residues at the cleavage site in the F gene. NDV genotype VII is the predominant genotype in poultry in the Middle East with most NDV isolates from wild birds additionally assigned to this genotype [51]. Additionally, a recent study conducted in Egypt on wild bird species, isolated velogenic NDV-VII from cattle egret and house sparrows collected from the vicinity of poultry farms with a history of NDV infection [30]. Such findings promote the hypothesis of the interplay of avian pathogens between wild birds and domestic poultry.

As NDV is an acute contagious disease affecting birds of all ages [52], the clinical observations of this study revealed that domestic birds including chickens, ducks, and turkey, infected with VNDV experience cyanosis of comb and wattles, greenish diarrhoea, respiratory and nervous distress, general septicaemia, and haemorrhages in the internal organs, with variable degrees of severity. These clinical and post-mortem observations correlate to the disease manifestation findings of birds infected with VNDV in other studies [10,53,54,55]. However, all examined migratory and non-migratory wild birds that were positive for NDV had no apparent clinical disease signs or post-mortem lesions, which are coherent with the findings recorded in previous studies related to ND infections in wild birds [30,56]. Domesticated and wild duck species are known to present varied clinical disease pattern under NDV infection, and isolation of pathogenic NDV has been gradually increasing in waterfowl [23,57,58,59,60]. The experimental infection of Muscovy ducks with NDV genotype VII led to only 5% mortality despite significant tracheal and cloacal shedding, however close contact chicken experienced severe symptoms with higher mortality rates, emphasizing the role of ducks as effective carriers of NDV [61]. In the current study, NDV as a single infection was isolated from 17 clinically diseased ducks and only one flock was co-infected with NDV and AIV. Additionally, VNDV-VII with AIV were identified in two out of six investigated duck farms in Egypt suffering from respiratory manifestation and high mortality [62]. Such variability of pathogenicity of NDV in waterfowl species emphasizes their potential role as effective carriers or reservoir hosts for NDV.

Phylogenetic analysis revealed that all sequences of the six examined strains of domestic poultry: NDV/Chicken/Egypt/NOR/ZU-NM76/2019, NDV/Chicken/Egypt/ALEX/ZU- NM93/2019, NDV/Chicken/Egypt/ALEX/ZU-NM97/2019, and NDV/Chicken Egypt/ALEX/ZU-NM99/2019, those from ducks: NDV/Duck/Egypt/DAK/ZU-NM09/2019 and NDV/Duck/Egypt/DAK/ZU-NM54/2019, and one from migratory wild birds: “NDV/Black-crowned_night_heron/Egypt/POR/ZU-NM85/2019” were velogenic and clustered together in class II within sub-genotype VII.1.1, with 98.013–100% nucleotide identity, complementary to findings from related studies [10,55,63,64,65]. Additionally, the strains were highly similar to sequences of NDV-GVII.1.1 isolates from neighbouring countries such as those in Israel, 2017 and China, 2011 (98.11–98.01%); genotype VII was the predominant genotype in China of NDV during 2000–2015 [66,67,68]. Field strains isolated in Israel since 2000 mainly belong to sub-genotypes VIIb, VIId, and VIIi of genotype VII (class II) [42,69,70]. These findings suggest that this genotype remains endemic and circulating in several countries, genetically evolving within respective regions, including Egypt. Furthermore, the pathogenicity of the six sequenced strains confirmed the high virulence with values of MDT (35.8–64) and ICPI (0.98–1.670); ICPI values above 0.7 are deemed virulent [31]. Coupled with the presence of a poly-basic cleavage site in the F protein, the sequenced strains were confirmed as highly virulent.

Few amino acid (aa) substitutions (N30S, T90A, N258S, and R480K) were found in the entire length of F protein in APMV-1 isolates from chicken, duck, and wild birds, as compared with each other. Such substitutions have been proposed to influence the antigenicity as well as a change in the fusion protein confirmation [10,60,71,72,73,74,75,76]. The substitution particularly in the surface-exposed aa residues in the signal peptide cleavage site of NDV F protein (position: 30) is considered to be highly variable as this region must undergo constant positive selection [71]. Orabi et al. [72] detected that the N30 residue appears to be conserved in all vaccine and Egyptian field strains clustered within genotype II; two isolates from this study “NDV/Duck/Egypt/DAK/ZU-NM09/2019” and “NDV/Chicken/Egypt/ALEX/ZU-NM97/2019” shared such residue. This suggested that some ND viruses of genotype VII in Egypt may induce a new selection profile based on mutation categories of field virus isolates in the presence of vaccines of different genotypes [72,73]. Three isolates (NDV/Chicken/Egypt/NOR/ZU-NM76/2019, NDV/Black-crowned_night_heron/Egypt/POR/ZU-NM85/2019, and NDV/Chicken/Egypt/ALEX/ZU-NM99/2019) showed the substitution R480K within a domain essential for virus fusion. NDV-VII isolated during 2016 in Egypt had a conserved amino acid residue (R480), but recorded a nearby substitution (N469S) that was hypothesised to induce high mortality among infected chickens [72]. Further study is required to investigate the viral fitness following such substitutions detected within the F protein of NDV of the isolates of this study.

The results of this study observed a predominant isolation of NDV-VII.1.1 with the velogenic characteristic, from domestic and wild birds, with a high degree of genetic similarity across distinct governates in Egypt. As such, the data may suggest an interlinking role contributing to the maintenance and transmission cycle of NDV between and within domesticated poultry and wild birds situated in Egypt. Previous literature highlights the prevalence of NDV in wild birds driving dissemination across the governates and coupled with this study’s data, a stronger epidemiological picture can be formed, complementing the role of wild birds aiding such maintenance but also suggesting their role in recurring outbreaks in poultry [77,78,79]. As such, further surveillance of NDV evolution and prevalence among migratory and free-living birds simultaneously with domestic avian species would significantly strengthen our understanding of NDV in Egypt. Such efforts are also critical for the implementation of appropriate protective vaccines as well as necessitating improved biosecurity systems to impede the virus transmission cycle and protect domestic poultry.

## Figures and Tables

**Figure 1 viruses-14-02244-f001:**
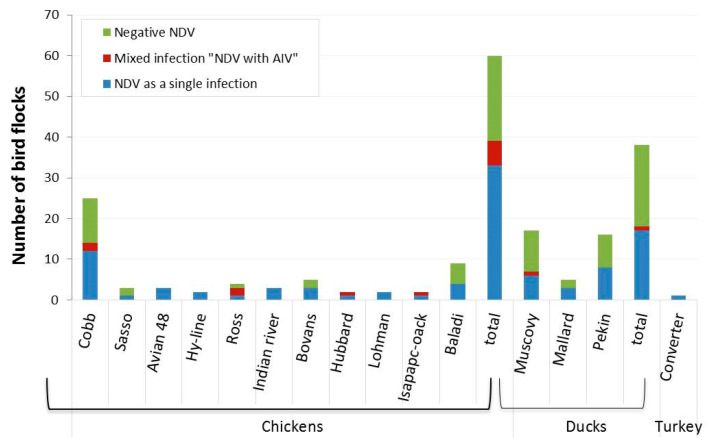
Domestic bird breeds in Egypt positive for NDV as a single natural infection (detected via NDV F gene-specific RT-PCR, blue bars) or mixed natural infection with NDV and AIV (detected via flu M gene-specific RT-PCR, maroon bars). The number of birds negative for NDV are indicated as green bars.

**Figure 2 viruses-14-02244-f002:**
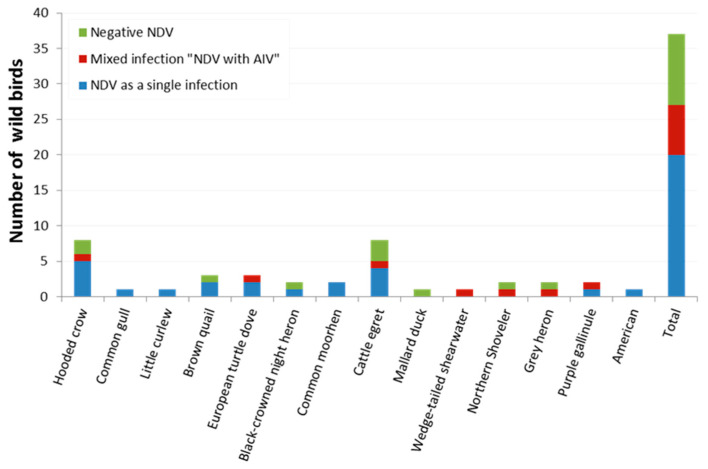
Wild bird breeds in Egypt positive for NDV as single natural infection (detected via NDV F gene-specific RT-PCR, blue bars) or mixed natural infection with NDV and AIV (detected by Flu-M gene-specific RT-PCR, maroon bars). The numbers of birds negative for NDV are indicated as green bars.

**Figure 3 viruses-14-02244-f003:**
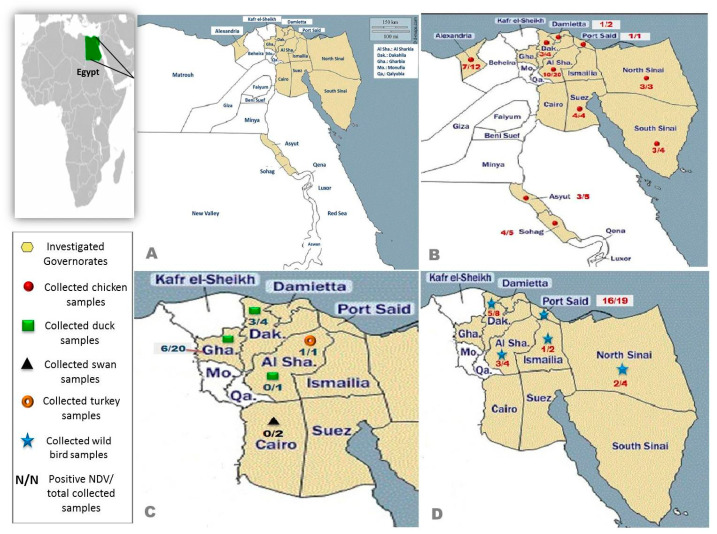
Geographical spread pattern of Egyptian NDV. (**A**) The Egyptian map shows the governorates of collected samples (yellow). (**B**) Distribution of tested chicken (red) samples. (**C**) Distribution of tested duck (green), turkey (orange), and pelican (black), samples. (**D**) Distribution of tested wild bird (blue) samples. N/N positive NDV to the total collected samples in each governorate.

**Table 1 viruses-14-02244-t001:** Details of collected samples from domestic birds in Egypt, during 2019–2020.

Species	Breed	No. of Flocks	Age/Day	Type of Poultry Flock	Location
Chickens	Cobb	25	23–45	Broiler	Sharkia, Dakahlia, Sohag, Alexandria, Assuit, North and South Sinai
	Sasso	3	40–310	Broiler/Layer	Dakahlia, Sohag, Damietta
	Avian 48	3	30–33	Broiler	Suez, South Sinai
	Hay-line	2	200	Layer	Dakahlia
	Ross	4	28–35	Broiler	Alexandria, North Sinai
	Indian river	3	32–48	Broiler	Suez, North Sinai
	Bovanes	5	210–252	Layer	Sharkia, Alexandria
	Hubbard	2	27–29	Broiler	Damietta, Port-Said
	Lehman	2	200–330	Layer	South Sinai, Alexandria
	Isapapcoack	2	100	Layer	Sharkia
	Baladi	9	20–40	Native	Sharkia, Sohag, Assuit
Ducks	Muscovy	17	25–360	Broiler/Breeder	Dakahlia, Gharbia
	Mallard	5	187–622	Broiler/Breeder	Dakahlia, Gharbia
	Pekin	16	30–700	Broiler	Sharkia, Dakahlia, Gharbia
Turkeys	Converter	1	47		Sharkia
Pelican	Pelecanus crispus	2	4320–5400		Cairo

**Table 2 viruses-14-02244-t002:** Details of collected samples from wild birds, including order, species, common name, location of capture, collection date, and number of captured birds.

Species	Common Name	Number of Birds	Collection Date (Month/Year)	Location
*Corvus cornix*	Hooded crow	8	4, 8, 10/2019	Sharkia, Port-Said, Dakahlia
*Larus canus*	Common gull	1	3/2019	Port-Said
*Numenius minutus*	Little curlew	1	3/2019	Port-Said
*Coturnix ypsilophora*	Brown quail	3	4/2019	North Sinai
*Streptopelia turtur*	European turtle dove	3	4, 10/2019	Port-Said, Sharkia
*Nycticorax nycticorax*	Black-crowned night heron	2	10/2019	Port-Said
*Gallinula chloropus*	Common moorhen	2	10/2019	Port-Said
*Bubulcus ibis*	Cattle egret	8	8, 10/2019	Port-Said, Ismailia, Dakahlia
*Anas platyrhynchos*	Mallard duck	1	7/2019	North Sinai
*Ardenna pacifica*	Wedge-tailed shearwater	1	3/2019	Port-Said
*Spatula clypeata*	Northern Shoveler	2	3/2019	Port-Said
*Ardea cinerea*	Grey heron	2	3/2019	Port-Said
*Porphyrio madagascariensis*	Purple gallinule (African swamphen)	2	3, 4/2019	Port-Said
*Anthus rubescens*	American pipit	1	4/2019	Port-Said

**Table 3 viruses-14-02244-t003:** Incidence of NDV among naturally infected domestic birds in Egypt.

Breeds	No. of Flocks	Detected Virus(es)			
		No of Positive NDV (%)	95% CI	No of Positive AIV (%)	95% CI
Chickens					
Cobb	25	14 (56)	34.9–75.6	4 (16)	4.54–36.08
Sasso	3	1 (33.3)	0.84–90.57	0	0
Avian 48	3	3 (100)	29.24–1.00	0	0
Hy-line	2	2 (100)	15.81–1.00	0	0
Ross	4	3 (75)	19.41–99.37	2 (50)	6.76–93.24
Indian river	3	3 (100)	29.24–1.00	0	0
Bovans	5	3 (60)	14.66–94.73	1 (20)	0.51–71.64
Hubbard	2	2 (100)	15.81–1.00	1 (50)	1.26–98.74
Lohman	2	2 (100)	15.81–1.00	0	0
Isapapcoack	2	2 (100)	15.81–1.00	1 (50)	1.26–98.74
Baladi	9	4 (44.4)	13.70–78.80	0	0
Total	60	39 (65)	51.60–76.87	9 (15)	7.10–26.57
Ducks					
Muscovy	17	7 (41.2)	18.44–67.08	3 (17.6)	3.80–43.43
Mallard	5	3 (60)	14.66–94.73	0	0
Pekin	16	8 (50)	24.65–75.35	1 (6.3)	0.16–30.23
Total	38	18 (47.4)	30.97–64.19	4 (10.5)	2.94–24.80
Turkeys					
Converter	1	1 (100)	2.50–1.00	0	0
Pelicans					
Pelican crispus	2	0	0	0	0

**Table 4 viruses-14-02244-t004:** Incidence of NDV among naturally infected wild birds in Egypt.

Species	Common Name	No. of Birds	Detected Virus(es)			
			No. of Positive NDV (%)	95% CI	No. of Positive AIV (%)	95% CI
*Corvus cornix*	Hooded crow	8	6 (75)	34.91–96.61	1 (12.5)	0.32–52.65
*Larus canus*	Common gull	1	1 (100)	2.50–1.00	0	0
*Numenius minutus*	Little curlew	1	1(100)	2.50–1.00	0	0
*Coturnix ypsilophora*	Brown quail	3	2 (66.7)	9.43–99.16	0	0
*Streptopelia turtur*	European turtle dove	3	3 (100)	29.24–1.00	1 (33.33)	0.84–90.57
*Nycticorax nycticorax*	Black-crowned night heron	2	1 (50)	1.26–98.74	0	0
*Gallinula chloropus*	Common moorhen	2	2 (100)	15.81–1.00	0	0
*Bubulcus ibis*	Cattle egret	8	5 (62.5)	24.49–91.48	1 (12.5)	0.32–52.65
*Anas platyrhynchos*	Mallard duck	1	0	0	0	0
*Ardenna pacifica*	Wedge-tailed shearwater	1	1 (100)	2.50–1.00	1 (100)	2.50–1.00
*Spatula clypeata*	Northern shoveler	2	1 (50)	1.26–98.74	1 (50) *	1.26–98.74
*Ardea cinerea*	Grey heron	2	1 (50)	1.26–98.74	1 (50) *	1.26–98.74
*Porphyrio madagascariensis*	Purple gallinule (African swamphen)	2	2 (100)	15.81–1.00	1 (50)	1.26–98.74
*Anthus rubescens*	American pipit	1	1 (100)	2.50–1.00	0	0
Total		37	27 (72.97)	55.88–86.21	7 (18.9)	7.96–35.16

(*) birds co-infected with NDV and AIV.

**Table 5 viruses-14-02244-t005:** The details of whole genome sequencing of NDV Egyptian strains of this study.

Name of Isolates	Breed	Locality	Descriptive Data					Sequencing	Accession Number
			Date	Age/Days	Flock Density	Mortality Rate/ 3 Days	Clinical Finding		
NDV/Chicken/Egypt/NOR/ZU-NM76/2019	Ross (Domestic)	North Sinai	03/2019	33	14,000	600	Respiratory signs, greenish diarrhoea, septicaemia	Approx. 99% genome coverage (103 nucleotides gap), Full F gene	OP219678
NDV/Chicken/Egypt/ALEX/ZU-NM97/2019	Ross (Domestic)	Alexandria	05/2019	30	5000	62	Respiratory signs, greenish diarrhoea, haemorrhages in caecal tonsils and proventriculus	Approx. 90% genome coverage, Full F gene	OP219679
NDV/Chicken/Egypt/ALEX/ZU-NM99/2019	Ross (Domestic)	Alexandria	05/2019	25	6000	85	Respiratory signs, greenish diarrhoea, haemorrhages in caecal tonsils and proventriculus	Approx. 99% genome coverage (150 nucleotides gap), Full F gene	OP219680
NDV/Duck/Egypt/DAK/ZU-NM09/2019	Mallard (Domestic)	Dakahlia	03/2019	300	3300	185	Nervous signs, greenish diarrhoea, congestion in parenchymatous organs	Approx. 99% genome coverage (153 nucleotides gap), Full F gene	OP219681
NDV/Black-crowned night heron/Egypt/POR/ZU-NM85/2019	Migratory (Wild Bird)	Port-Said	09/2019	N/A	N/A	N/A	Apparently healthy without PM lesions	Approx. 90% genome coverage, Full F gene	OP219682
NDV/Duck/Egypt/DAK/ZU-NM54/2019	Muscovy (Domestic)	Dakahlia	03/2019	60	4000	113	Greenish diarrhoea, septicaemia	Approx. 70% genome coverage (3918 nucleotide gap), Partial F gene	OP219683

## Data Availability

Participant data will be made available upon reasonable requests directed to the corresponding author. Proposals will be reviewed and approved by the investigator and collaborators on the basis of scientific merit. After approval of a proposal, data can be shared through a secure online platform after signing a data access agreement.

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
