# Peer review of "Newcastle Disease Genotype VII Prevalence in Poultry and Wild Birds in Egypt"

_viruses, 2022, doi:10.3390/v14102244_

Round 1

Reviewer 1 Report

Eid et al describe the prevalence of virulent APMV-1, commonly known as Newcastle Disease Virus (NDV), in circulation throughout Egypt in domesticated poultry and within wild-birds. A high number of samples were analysed with detection of APMV-1 was observed in a high proportion of poultry and wild-bird samples, along with a number of NDV/AI mixed infections. They detected virulent genotype VII in poultry, as has been seen previously, but also in a wild-bird sample. However, this single observation of a genotype VII virus in a wild-bird sample has led to a lot of speculation of the role of wild birds in spreading NDV.

Major Points

Introduction

Line 34 – State “ND was first identified in 1948.” Do you mean in Egypt? If so, please make this point clear

Materials and Methods

Table 1 – Stated that the species is swan, but breed is Pelican Crispus. This refers to a pelican not a swan, please amend.

Results

3.2 Virus identification – I found this whole section confusing to read, with the figures of little help to decipher what was been described in the text. The graphical information may be better suited to a table, so can be easily cross-referenced with the text.

Figure 1 and Figure 2 figure legends have been mixed up

Line 174 – states that 20 isolates were selected for whole genome sequencing, but only 6 viruses were selected. Why was this? Surely the conclusions would be much more robust if 20 genomes, or F-gene sequences were included.

As a further comment, in line 155 it is stated that there 85 samples positive for APMV-1 by RT-PCR analysis. Could more of these samples be sequenced, even if it is by Sanger sequencing and is just the cleavage site to determine the prevalence of virulent APMV-1 (NDV) within Egypt, as it could be that some of these sequences relate to recently vaccinated domestic poultry.

Why was only a single wild bird sequenced? To make the argument that wild-birds play a role in the spread of genotype VII.1.1 APMV-1, this would need to have been observed in more than one sample.

Lines 208 and 209 – Uses VII.1.1.j This is a mixture of Dimitrov and Diel nomenclature, please use the most up to date nomenclature (Dimitrov 2019). Please also be consistent with nomenclature throughout the manuscript

Discussion

Lines 232 – 235 Stated that all wild-birds sampled were positive for VNDV-GVII. Firstly, what do you mean by VNDV-GVII? VNDV isn’t described until line 262, please make this clear. Secondly, is this based on the single sample from the Black-crowned night heron? If so, you need more evidence than a single sample to make this claim.

Line 286 – States that 7 samples were examined, and all were VII.1.1, yet only 5 are commented on in this study. Why were these two samples not included. Please either include in the analysis, or remove reference to these samples if the sequence was not of a good enough quality

Line 297 - 298 – “close phylogenetic relationship between circulating NDV strains in migratory birds and those detected in domestic poultry.” Again, is this due to the single wild-bird sample that was observed? Again, the genotype VII isolate needs to be observed in more than one sample to give any weight to this claim.

Where might the wild-birds have initially contracted this genotype VII virus, as previously, nearly all cases have been described in domestic poultry.

Supplementary Table 1-Could the genotype VII samples be placed next to the samples from Egypt so it is easier to observe the relative homology

Supplementary Figure 3 – The most recent F-gene sequence included is from 2016. Are there anymore recent F-gene sequences from Egypt (e.g. those examined in refs 75 and76) that could be included to examine temporal relationship of these sequences?

Reviewer 2 Report

In this manuscript, the authors investigate the NDVs from domestic versus migratory and non-migratory birds in Egypt and evaluated the potential role of wild birds in NDV transmission to the poultry industry. All five NDV in this study were genotype VII.1.1 and the wild bird isolate shared 94.15-96.19% of nucleotide sequence indentity in F gens with NDV isolated from chickens and ducks. The authors suggest the wild bird play an potential role in the dissemination of NDV in poultry populations across Egypt. In general, this manuscript needs to include more supporting data and analyses of the obtained results. I would suggest that the authors are willing to re-write the manuscript and remove all statements that are not supported by real data and focusing on the epidemiology and molecular characterization of  NDVs circulated in poultry and wild birds in Egypt.

Major comments:

1. In the title, inadequate evidence from the work done here supports the statement “Newcastle Disease genotype VII outbreaks in poultry across Egypt are associated with enzootic prevalence of the virus in wild birds”.

2. The discussion need a completed revision for style and contents. It need to be shortened and focused on the contributions of this manuscript, instead of reviewing the literature.

3. The authors just abtained 5 full F gene sequences from these isolates. It is the opinion of this reviewer that more F gene sequences should be collected and analyse.

Minor comments

Line 27: The abbreviation of Avian Orthoavulavirus 1was not APMV-1.  

Line 31:  Not all class I NDVs were avirulent viruses.

Line 34: ND was first identified in 1927. In 1948 in Egypt?

Line 73: Provide Clinical findings in table 1.

Line 84: Which were migratory and non-migratory wild birds?

Line 88: µl should be µL, and others in the manuscript.

Line 139: The authors should explain why the GTR nucleotide substitution model was applied, such as perform model selection.

Line 163: The legend of  “Figure 1” and “Figure 2” is confused.

Line 250-254: Serval studies have reported that virulent NDVs isolated in wild birds and ducks. The authors should reword this sentence.

Reviewer 3 Report

Amal A.M. Eid and co-authors have performed a study to investigate the prevalence of NDV in poultry and wild birds in Egypt. They have isolated six NDV strains from collected samples and the virus whole genome were amplified and sequenced. Furthermore, they constructed phylogenetic tree for phylogenetic analysis. The author demonstrated that class II genotype Ⅶ.1.1 strain was the epidemic strain. Sustained NDV surveillance in poultry and wild birds are important to control ND, rendering the manuscript of interest. However, there are some flaws that need to be addressed.

Major comments:

Comments and suggestions:1. Some descriptions are wrong or inaccurate, such as Lines 17~18, based on the presence of polybasic amino acids (RRQRF) at the F gene cleavage site ? this sentence should be changed to “based on the presence of polybasic amino acids (RRQRRF or RRQKRF) at the F protein cleavage site”. line 29, NDV encoded at least six different proteins. Please check the full manuscript carefully. Line 286, “7 examined strains of domestic poultry…”, but table 5 only show information of six strains. Furthermore, line 286~292 states the 7 examined strains belongs to class II within sub-genotype VII.1.1, but the Suppl. Figure 1 only have five isolated strains. Line2 19~190, “The homology analysis of the five viral genome sequences demonstrated that the F genes shared very high nucleotide sequence identity of 98.04~100% (Table S1).”, but table S1 showed the nucleotide sequence identity of the five viral genomes were 98.013~100%, please check.

Comments and suggestions:2. From the manuscript, we found that NDV isolation rates were relatively high in poultry and wild birds. With such a relatively higher isolation rates, the authors need to give a detailed description of the vaccination of these domestic flocks in materials and methods. Did these flocks vaccinated with NDV vaccine or not ?

Comments and suggestions:3. As the author state all the six NDV are virulent strains according to the cleavage site of F protein, but sometimes wild birds origin NDV strains with polybasic amino acids (RRQRF) at the F protein cleavage site show low virulence for chickens. So, authors need to test the virulence of the six isolated viruses by ICPI. So we can acquire more valuable information about viruses.

Comments and suggestions:4. The author declare that they identified 85 NDV positive samples through HA and RT-PCR, in which 71 samples were finally verified as single NDV infection. Why were only 20 NDV strains isolated? Among the 20 NDV isolates used for whole genome sequencing (WGS), why were only 6 viruses selected for sequence analysis? If more strains were choose for genome amplification and sequence analysis, the authors may be identified other genotypes of NDV.

Comments and suggestions:5. The author use SISPA to amplify the whole genome of the six strains, but only acquired five full F gene sequences. The above result suggest that SISPA was not superior than conventional RT-PCR to acquire even complete F gene. And the complete genome was not used for further analysis. Why the authors choose SISPA method rather than conventional PCR to amplify NDV whole genome or F gene.

Round 2

Reviewer 1 Report

Eid et al have revised their original document “Newcastle Disease genotype VII outbreaks in poultry across Egypt are associated with enzootic prevalence of the virus in wild birds.” I thank the authors for addressing the comments and queries appropriately. However, the fact still remains that the title of the paper and therefore the whole premise is based on the single observation of a genotype VII virus in a wild bird, albeit linked phylogenetically to other Egyptian APMV-1 from domestic poultry and ducks. Until more sequences are obtained from wild birds which demonstrate the presence of Gen VII APMV-1 is commonly observed throughout Egypt, rather than the more common Class I or avirulent Class II APMV-1’s, I cannot see how the data justifies the title.

Within the article, it is stated that APMV-1 was isolated from eggs from 85 swabs, of which 27 were from wild birds. Therefore, can the RNA be re-extracted and examined via Sanger sequencing to confirm at least the presence of virulent APMV-1 (NDV) rather than carrying out WGS? This would at least answer the question around virulence of the strains in the wild-bird samples, and also the presence of vaccine strains within domesticated birds.

Author Response

Please see our replies to the comments 1&2 of reviewer-1:

(1): Eid et al have revised their original document “Newcastle Disease genotype VII outbreaks in poultry across Egypt are associated with enzootic prevalence of the virus in wild birds.” I thank the authors for addressing the comments and queries appropriately. However, the fact still remains that the title of the paper and therefore the whole premise is based on the single observation of a genotype VII virus in a wild bird, albeit linked phylogenetically to other Egyptian APMV-1 from domestic poultry and ducks. Until more sequences are obtained from wild birds which demonstrate the presence of Gen VII APMV-1 is commonly observed throughout Egypt, rather than the more common Class I or avirulent Class II APMV-1’s, I cannot see how the data justifies the title.

Reply: The title has been amended to “Newcastle disease genotype VII prevalence in poultry and wild birds in Egypt”. The focus of the paper has altered to observation to better reflect the data and the context of the study. Further resources and time would be required to expand the data to conclusive statements, as pointed out by the reviewers, and thus, the remarks of the study state observational prevalence.

2: Within the article, it is stated that APMV-1 was isolated from eggs from 85 swabs, of which 27 were from wild birds. Therefore, can the RNA be re-extracted and examined via Sanger sequencing to confirm at least the presence of virulent APMV-1 (NDV) rather than carrying out WGS? This would at least answer the question around virulence of the strains in the wild-bird samples, and also the presence of vaccine strains within domesticated birds.

Reply: We agree that this method would benefit the conclusions/observations of this study. However, due to the limited available time and resources, we have been unable to perform the sequence analysis of all the isolates collected in this study. to address this shortcoming, we have included the ICPI values of WGS strains which showed high virulence with the presence of the polybasic cleavage site, concluding that these strains are indeed deemed virulent. Also, the study does include the vaccine status of the domesticated birds, which were negative for any NDV-antigen vaccine, so all positive samples were considered to be naturally infected.

Reviewer 2 Report

Limited changes were made in the revised manuscript. I suggest authors that the changes should been marked in red and showed the location in the revised manuscript in replies.

1. Although the high incidence (72.9%) of NDV among wild birds which is more than those in domestic birds (57.4%) during 2019-2020 in Egypt, these also can not  supports the statement “Newcastle Disease genotype VII outbreaks in poultry across Egypt are associated with enzootic prevalence of the virus in wild birds” This just the deductions of the authors. I would suggest that this deducation should been discussed in the discussion  and revised the title.

2. In the first round of review, I suggest that the discussion need a completed revision for style and contents, and need to be shortened and focused on the contributions of this manuscript, instead of reviewing the literature. However, I can not find the changes in the revised manuscript.

Author Response

Replies to comments 1 &2 of Reviewer-2

(1): Although the high incidence (72.9%) of NDV among wild birds which is more than those in domestic birds (57.4%) during 2019-2020 in Egypt, these also can not support the statement “Newcastle Disease genotype VII outbreaks in poultry across Egypt are associated with enzootic prevalence of the virus in wild birds” This just the deductions of the authors. I would suggest that this deducation should been discussed in the discussion and revised the title.

Reply: The title has been amended to “Newcastle disease genotype VII prevalence in poultry and wild birds in Egypt”. The focus of the paper has altered to observation to better reflect the data and the context of the study. Further resources and time would be required to expand the data to conclusive statements, as pointed out by the reviewers, and thus, the remarks of the study state observational prevalence.

(2): In the first round of review, I suggest that the discussion need a completed revision for style and contents and need to be shortened and focused on the contributions of this manuscript, instead of reviewing the literature. However, I can not find the changes in the revised manuscript.

Reply: 

As suggested, the discussion has been shortened and adjusted to increase the focus on the results of this study.

Reviewer 3 Report

The authors have made modifications according to the requested major revisions. And they have added a clear point-by-point rebuttal to the revised manuscript, which made the reviewing process easier. But there are still some parts need to be revised.

Minor compulsory revisions:

1.The authors did not add experimental data about the virulence of the six isolated viruses as requested. The authors claim in the result of revised manuscript that “The F protein cleavage site sequence of all five isolates contained polybasic residues, (RRQXRF), where X is R or K, confirming their high virulence pathotype”, and they stated they investigated the pathogenicity of isolated viruses among different species in another project in their rebuttal letter, so they should give some evidence (like ICPI value) that the viruses are indeed highly virulent or rephrase this part.

2. The author has corrected most of the obvious wrong or inaccurate descriptions in the revised manuscript, but there are still some errors, such as line 164~165. The concentration of chicken red blood cell in HA assay should be 0.5% to 1% (V/V) , not 10%. Please check the whole manuscript carefully.

Author Response

Replies to the comments 1&2 of Reviewers-3 

(1): The authors did not add experimental data about the virulence of the six isolated viruses as requested. The authors claim in the result of revised manuscript that “The F protein cleavage site sequence of all five isolates contained polybasic residues, (RRQXRF), where X is R or K, confirming their high virulence pathotype”, and they stated they investigated the pathogenicity of isolated viruses among different species in another project in their rebuttal letter, so they should give some evidence (like ICPI value) that the viruses are indeed highly virulent or rephrase this part.

Reply: The “2.8. Pathogenicity indices” chapter has been included in the “Materials and Methods”. This addresses the ICPI values and complements the presence of a polybasic cleavage site, confirming the virulence of the sequenced strains.

(2): The author has corrected most of the obvious wrong or inaccurate descriptions in the revised manuscript, but there are still some errors, such as line 164~165. The concentration of chicken red blood cell in HA assay should be 5% to 1% (V/V) , not 10%. Please check the whole manuscript carefully.

As advised, the text has been amended accordingly for the above note.

Round 3

Reviewer 1 Report

Thank you to the authors for the modifications made to the document. I am now happy with the manuscript